# Nitrate Management Discourses in Poland and Denmark—Laggards or Leaders in Water Quality Protection?

**Emilia Noel Ptak** [1,*], **Morten Graversgaard** [1,2], **Jens Christian Refsgaard** [3] and **Tommy Dalgaard** [1]

1 Department of Agroecology, Aarhus University, Blichers Alle 20, 8830 Tjele, Denmark; morten.graversgaard@agro.au.dk (M.G.); tommy.dalgaard@agro.au.dk (T.D.)
2 Department of Geosciences and Natural Resource Management, University of Copenhagen, Øster Voldgade 10, 1350 Copenhagen, Denmark
3 Geological Survey of Denmark and Greenland (GEUS), Øster Voldgade 10, 1350 Copenhagen, Denmark; jcr@geus.dk
* Correspondence: eptak@agro.au.dk

**Abstract:** The most significant source of nitrate pollution in the European Union (EU) is attributed to agricultural activities, which threaten drinking water, marine, and freshwater resources. The Nitrates Directive is a key feature of the Water Framework Directive (WFD), which seeks to reduce nitrate pollution from agricultural sources. Yet, weak compliance by Member States (MS) diminishes the legitimacy of the EU environmental acquis and undermines efforts to achieve environmental objectives. This study examines the nitrate management discourse in Poland to identify influencing factors that impact governance capacity and overall compliance performance. The empirical investigation is based on nine stakeholder interviews, three written correspondences, and a literature review that collectively comprise an evaluation study. A comparison in governance approaches between Poland and Denmark provides a calibration in assessing performance respective to another MS. The findings categorize both Poland and Denmark as "laggard" in WFD compliance. This case contributes new insights in identifying 6 enabling and 13 constraining factors affecting the ability of MS to fulfill their implementation duties. The findings demonstrate that divergent stakeholder views based on historical and cultural norms require a differentiated approach tailored to domestic conditions for effective fulfillment of the objectives set forth in EU environmental legislation.

**Keywords:** Nitrates Directive; Water Framework Directive; implementation performance; leaders and laggards; policy design; agriculturally induced water pollution; comparative governance approaches

## 1. Introduction

The European Union has sought to ameliorate agriculture's impact on the environment through the establishment of various policies. The Nitrates Directive (ND) (91/676/EEC) is the main legislative framework that seeks to reduce nitrate pollution from agricultural sources to protect Europe's waters. Member States are responsible for nitrate management by designating areas sensitive to nitrate pollution, establishing measures, and monitoring water quality [1]. The Nitrates Directive is a central instrument of the Water Framework Directive (WFD) (2000/60/EC), which is based on an integrative river basin management approach. The overall WFD objective is to achieve "good ecological and chemical status" of all European waters to safeguard the integrity of natural ecosystems, human health, water supply, and biodiversity [2]. In recognition that water issues are transboundary and interconnected, a holistic policy approach is necessary for the sustainable management of Europe's water resources [3,4]. Thereby,

coordination amongst multiple policy domains is an important element for the functioning of an integrated water governance system. As a crosscutting issue, agricultural diffuse pollution links the ND and WFD in pursuit of the common goal to protect surface and ground water quality. Thus, compliance performance of the WFD is greatly influenced by the ability of Member States (MS) to implement the ND in successfully addressing agricultural diffuse pollution.

Management of agricultural diffuse pollution is a complex issue [5], intersecting at the nexus of environmental, agricultural, and water policies. Agricultural diffuse pollution constitutes a "wicked problem" [6–9], involving a multitude of stakeholders, operating at different levels within a diverse range of contexts [10] that collectively contribute to the tragedy of the common scenario [11]. Thus, the scope and scale of the issue calls into question the ability of traditional centralized governance structures to adequately manage this complex environmental challenge. Meeting the objectives of the WFD requires organizational restructuring of institutions and administrative arrangements, which remains a persistent challenge, as evidenced by ongoing issues of governance fragmentation within the majority of MS [12]. The management of Europe's waters is a particularly contentious area of environmental governance, as evidenced by Member States experiencing difficulties in fulfilling the requirements of the Nitrates Directive [13,14] and the WFD [15–18].

During the past decade, the field of environment has consistently ranked highest in the total number of open infringement cases, which contributes to weakening overall compliance and the ability of the EU to achieve environmental objectives [19–22]. Weak compliance diminishes the legitimacy of the EU environmental acquis and renders efforts to sustainably manage common pool resources (CPR) [23] ineffective. The trend of a high number of infringement cases for the area of environment does not express uniformity for an overall categorization of EU environmental policy performance. Rather, implementation performance is a nuanced and dynamic process that shifts based on an interplay of factors situated within a particular temporal and spatial context. Thereby, a MS categorized as lagging behind in compliance with a particular environmental directive, may demonstrate a leadership role in addressing a different environmental challenge. Yet assessments are often posited as static categorizations that a MS fits into [24]. Implementation performance is broadly applied to cover an entire country, yet outcomes can vary within a country based on a confluence of influencing factors in shaping governance capacity within local level contexts [7].

Themes central to assessing effective governance of the WFD, identified by Wiering et al. [12], include (1) integration and fragmentation, (2) source-and-effect-based measures and (3) intricacies of knowledge production. A number of studies explore MS implementation performance of the WFD in light of these themes to better understand the conditions and processes that give rise to particular compliance outcomes [5,11,25,26]. Empirical investigations of management discourses within local landscapes can demonstrate implementation performance as a social practice and provide compelling case studies to understand the conditions that lead to a particular compliance outcome [27–30]. The point of departure of the study is to identify influencing factors that shape the nitrate management discourse situated at the intersection with the WFD and the Common Agricultural Policy (CAP) in Poland. The study identifies underlying influencing factors that contribute to shaping particular outcomes of the aforementioned themes central to governance of the WFD. The study focuses on theme 1 in understanding how an interplay of factors contributes to governance structures and ensuing social arrangements determining actor roles and responsibilities. In addition, a comparison of governance capacity between Poland and Denmark is made to calibrate implementation performance and discern wider lessons in relation to the WFD governance themes.

A comparative analysis of the nitrate management discourses in Poland and Denmark is relevant as the two countries are responsible for the reduction of nutrient loading to the Baltic Sea (see Figure 1). Both states dedicate more than half of their land area to agricultural activities, with a significant portion for livestock production, which produces a high amount of nitrate pollution. The average farm holding size is approximately 11 ha in Poland and 67 ha in Denmark [31], demonstrating a large variance in the agricultural systems. Denmark has classified its entire territory as a nitrate vulnerable zone

(NVZ), as all surface waters drain to marine waters. Poland recently designated its entire territory as NVZ and 98% of its surface waters drain to the Baltic Sea [32]. The domestic situations differ between Poland and Denmark, yet both countries face considerable pressure to reduce their respective nutrient loadings to coastal waters. The European Commission, regarding improper implementation of the Water Framework Directive [33–36], has referred both states to the Court of Justice of the EU. Thus, there is a shared responsibility for both states to improve compliance performance of the Water Framework Directive through better nitrate management.

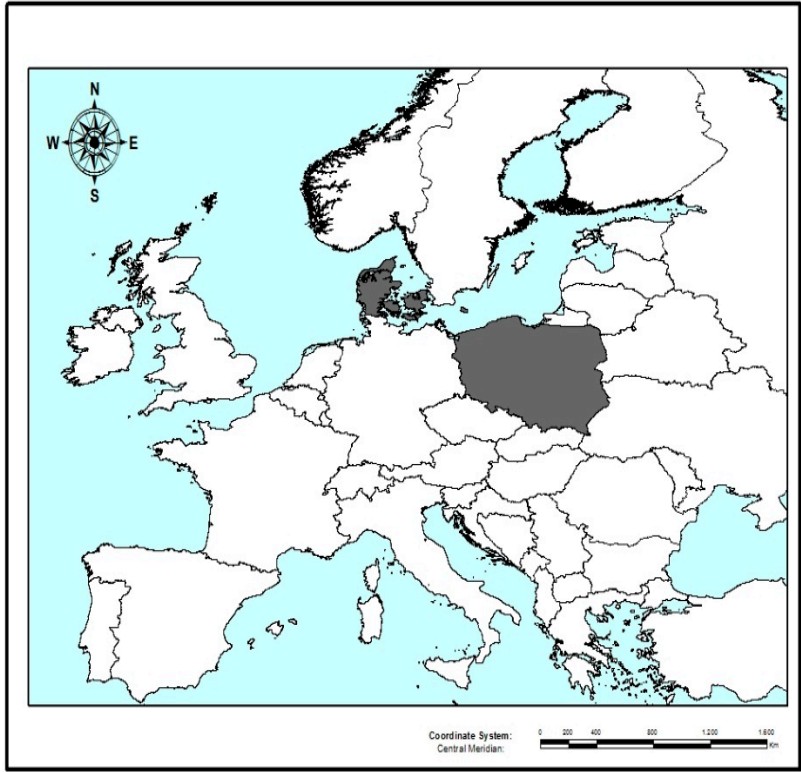

**Figure 1.** Map of Europe highlighting Poland and Denmark (ArcGIS®Esri, 2016).

This paper seeks to address the existing research gap concerning that evaluations of compliance performance are focused on transposition and application at the national level, neglecting that the regional and local levels are the places where implementation is realized in practice. The objective is threefold: (1) identify the enabling and constraining factors that shape implementation outcomes and contribute to a particular compliance performance categorization; (2) assess to what extent the local empirical investigation demonstrates coherence with national level compliance performance assessments; (3) make a comparative evaluation of Poland and Denmark to examine the dynamism of implementation discourses. Overall, the study provides a holistic approach of twinning local and national-level assessments to examine if there is convergence or divergence of implementation performance assessments at different scales. The study provides a more nuanced understanding of the intricate factor interplay that shapes implementation outcomes. In doing so, one can better anticipate a Member State's governance capacity to effectively comply with environmental Directives.

## 2. Theoretical Framework

Nitrate management is examined through the lens of a theoretical framework composed of four approaches that categorize compliance performance. Within the academic field of implementation performance assessment of European Union Directives, the four theories of leaders v. laggards [37–41], minimalist v. maximalist [42], worlds of compliance [43–45], and fit v. misfit [24,46–48] are well

recognized (Table 1). The field of EU compliance studies lacks an all-encompassing theoretical approach that takes account of diverse implementation patterns [48], and thereby offers "limited explanatory capacity" [43]. Further, a majority of these theories focus on national-level transposition, which subsequently neglects the dynamism and complexity of implementation performance taking place within local landscapes. In recognition that there is not one single determining factor of compliance performance [44], a comparative approach of the theories is taken to develop a more robust framework and to better provide an accurate picture of the implementation narrative taking place within a particular Member State.

**Table 1.** Macro-level theories on EU Member State Directive implementation compliance.

| Theoretical Approach | References | Non-Compliant | Compliant |
|---|---|---|---|
| Leaders or Pioneers (front-runners) Laggards or Stragglers (foot-dragging) Legal implementation "The way norms and standards are legally formulated and regulated in national law" [40] (p. 220) | [37–41] | Laggard or foot-dragging Rationale: unambitious national practices. Lenient and lack of enforcement of measures facilitate race to the bottom in terms of lowering policy performance and standards. | Leader or front runner Rationale: ambitious national practices. Strict interpretation and enforcement of measures. Pioneer through the initiation of innovative policy instruments, role model to other states who seek to emulate, lead and set advanced regulatory trends. |
| Minimalist v. Maximalist Policy approach taken in terms of formal compliance; to what extent legislation is adopted in terms of instituting minimal requirements or seeking to do more (maximal) | [42] | Minimalist Rationale: hands off approach, foot dragging with designation of NVZs. Low percentage of territory designated. | Maximalist Rationale: designate entire territory as NVZ. Perceived as "pioneer" or "leader" of EU environmental policy. |
| Worlds of compliance Worlds discerned regarding to what degree compliance is observed: 1. world of law observance 2. world of domestic politics 3. world of transposition neglect 4. world of dead letters | [43–45] | World of dead letters Rationale: pattern of pick-and -choose transposition and enforcement lacking. | World of law observance Rationale: pattern of compliance culture including enforcement. |
| Fit v. Misfit Goodness of fit Match v. Mismatch Success of EU policy implementation contingent upon level of "fit" with existing institutional structures and practices. | [24,46–48] | Misfit, Mismatch Rationale: rivalries among Ministries, which gives evidence to lack of cooperation and coordination. Further, lack culture of trust. | Fit, Match Rationale: taking an integrated approach as evidenced by entire territory designation as NVZ. Policy congruence present between domestic and EU levels. |

Taking a uniform approach to implementation performance neglects variance of conditions at the local level [24,40]. Thus, local cases may not fit constructed national meta-narratives of compliance performance [42,49]. The dissonance of the premise of uniform compliance behavior and analyzing performance at a particular level reveals a gap in the research field of evaluating environmental policy performance. In doing so, inaccurate categorizations of Member State compliance may be made [41]. Thus, it is important for monitoring purposes to analyze how implementation is realized in practice at the local level and further, to check for consistency with documentation of formal compliance submitted to the European Commission. The present study seeks to address the gap by offering an assessment that focuses on local level implementation situated within the larger context of governance capacity. In doing so, a more nuanced assessment of Member State compliance performance is provided for Poland. Governance capacity entails the ability of a state to mobilize and deploy organizational resources for effective and efficient regulation [50]. Determinants of effective governance capacity vary depending upon the context [51], and, therefore, broadly refers to the set of enabling governance conditions that empower stakeholders to address complex challenges [52,53]. For the scope of the research study, governance capacity centers on the ability of a MS to shift operational conditions of institutional and administration configurations to address agricultural diffuse pollution. Thereby, effective governance capacity is determined by the ability of MS to restructure institutional cultures that facilitate social norms and interactions based on the participatory approach of the WFD to realize an integrated water basin management governance system. The following framework (Table 1) is

utilized for the analysis categorizing Poland's implementation performance of the nitrate management discourse, along with the comparative evaluation with Denmark. The framework follows with detailed descriptions of the four theoretical approaches.

*2.1. Leaders and Laggards*

The leaders and laggards categorization ranks state performance according to regulation implementation records. States are considered to be leaders if they exhibit compliant behavior towards EU rules and directives are fully implemented in a timely manner. A "leader" or "front-runner" connotes a state that is positioned in front and influencing the direction of how a particular policy is pursued. Leader states can have a positive impact on the overall success of a policy by pulling up the performance standard of other states in facilitating a race to the top through their ambition and commitment to comply [38,41]. "Laggard" or "foot-dragging" connotes a state that is positioned behind in comparison to the performance of other states [37]. Non-compliance is exhibited by minimal or incomplete implementation of directives due to issues of institutional incompatibility, along with the lack of financial and administrative resources. Ranking implementation performance is useful in terms of discerning patterns of implementation performance among Member States and further anticipating how states will act in relation to future directives.

*2.2. Minimalist Versus Maximalist*

The minimalist versus maximalist approach refers to what extent European Union legislation is transposed into national law by Member States [42]. The approach offers a behavioral explanation regarding prevailing attitudes and interests of a state that lead to a particular interpretation of EU policy implementation. The maximalist approach is defined by strict and complete transposition, while the minimalist approach is defined by lenient and incomplete transposition. The maximalist approach connotes front-runner and leader categorizations. The minimalist approach connotes foot-dragging and laggard categorizations [42]. The focus is on the transposition stage of the implementation process, rather than the application and enforcement stages. In this sense, compliance is confined to evaluating state performance at the national level.

*2.3. Worlds of Compliance*

The worlds of compliance approach offers a typology of state behavior in relation to what extent compliance is an overriding national interest that the state pursues [43,44]. The theory explains a range of implementation patterns exhibited by Member States in relation to EU directives. There are four categories within the worlds of compliance: the world of law observance is explained by a culture of compliance, the world of domestic politics by political factors where domestic preferences have priority, and the world of neglect by administrative factors marked by incompatibility and inaction. A fourth typology called the world of dead letters was added to account for the unique position of the Central and Eastern European Countries (CEECs) and is characterized by a lack of administrative capacity in hampering implementation processes [44,45].

The world of law observance demonstrates MS compliance taking precedent over domestic concerns. Following EU law is part of a wider culture of compliance. Interpretation is ambitious in achieving EU objectives. Transposition takes place in a timely and correct manner. Additionally, domestic institutional arrangements and the administrative culture are well organized and have sufficient resources to support practical application, enforcement and monitoring of the EU legislation at multiple levels.

The world of domestic politics manifests as privileging domestic interests over EU legislation. Transposition depends on how well the EU measures match with domestic preferences, existing institutional arrangements, and administrative culture. If there is a conflict of interest, domestic concerns are prioritized.

The world of transposition neglect is categorized by compliance not being a goal that the MS aims to fulfill. MS in this category are characterized by inertia and do not recognize their transposition obligations. When transposition does take place, the minimal requirements are met, which fit within existing arrangements and structures. Application and enforcement are often lacking, due to negligence.

The world of dead letters refers to MS that demonstrate dissonance in compliance, depending on the stage of implementation. The legal transposition stage is marked by compliance, while the practical application and enforcement stages are marked by non-compliance. Lack of institutional capacity, weaknesses in civil service systems, and a traditional administrative culture are the main obstacles leading to the impediment of practical implementation [44].

### 2.4. Fit Versus Misfit

The fit versus misfit theory examines the effects of Europeanization on domestic institutional arrangements. Compliance performance is gauged according to what extent the transposition of EU legislation fits within the existing domestic institutional and administrative structures of Member States. If there is a high degree of fit, then transposition will follow a smooth process of clearly defined objectives, deadlines, and delegated responsibilities, and the legislation will be fully implemented [46,47]. Member States face adaptation pressure if there is a high degree of misfit, as the transposition process will be fraught with problems that hinder implementation, including a mismatch of expectations, institutional arrangements, and administrative culture, along with a lack of capacity to correct the misalignments. Implementation is therefore minimal, delayed, or incomplete. To improve the level of fit, state institutions are structured based on criteria of compatibility with EU expectations of effective governance to achieve Directive objectives. While the "fit versus misfit" [24], epithet is useful to compare policy approaches, it is more relevant to view it as a "process of framing", in recognition that implementation is marked by a high degree of institutional interplay. Compliance constitutes a shifting process producing a particular set of arrangements at a specific time and place [24]. Hence, implementation practices reflect a high degree of variance of fit and misfit in determining the most appropriate institutional response for a particular context.

## 3. Research Methodology

### 3.1. Case Study

The Polish study site 1 is situated within the Kocinka river catchment in the Silesia Province (Województwo śląskie), of Southern Poland (see Figure 2). Agriculture is the dominant land use feature, accounting for 54% of the area. Interviews were conducted with farmers residing within the catchment and with agency officials in the city of Częstochowa, which lies South of the river catchment. Study site 2 consisted of interviews conducted with an agency official and an agricultural advisor in the Kujawy-Pomerania Province (Województwo kujawsko-pomorskie). The map (Figure 2) highlights the two site areas of the study.

### 3.2. Evaluation Study

The qualitative research is comprised of an extensive literature review, content analysis of key stakeholder interviews, and identification of influencing factors that collectively comprise a trifecta of components for the implementation evaluation study. A comparison of nitrate management discourses between Poland and Denmark is made to provide a calibration in assessing performance respective to another Member State.

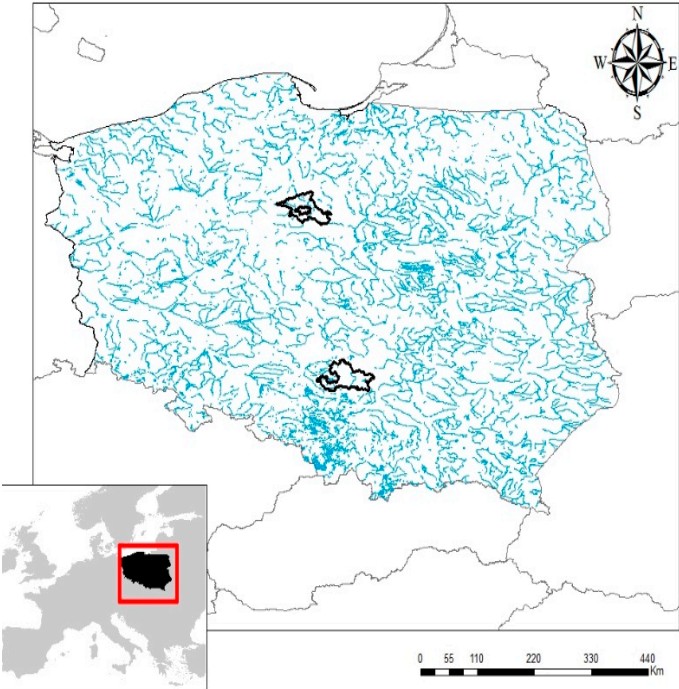

**Figure 2.** Study sites 1 (located in lower half of map) and 2 (located in upper half of map), along with system of major rivers in Poland (ArcGIS®Esri, 2016).

### 3.2.1. Literature Review

An extensive literature review on theories of Member State compliance, European Union water and agri-environmental policies, along with publications on water management pertaining to the country case studies, was made to evaluate the nitrate management discourses. The literature review included peer-reviewed journal publications, legislative documents, agency materials, conference papers, research studies, presentations, statistics, surveys, and historical documents.

### 3.2.2. Interviews

To gain a more nuanced understanding of the interplay of factors operating within the nitrate management discourse in Poland, interviews were conducted with stakeholders corresponding to different governance levels. Understanding the first-hand experiences and practices of key actors provides an insider perspective regarding the most relevant and current issues, as agricultural development is changing rapidly in Poland [54], and the existing body of literature does not keep pace with the rapid changes.

Structured and semi-structured interviews were conducted with key stakeholders during two field visits in Poland in May and July 2016. Structured interviews stipulate that the question set is established prior to the interview and is appropriate to utilize for a formal setting. The semi-structured approach is appropriate to employ when pre-existing theory can guide the research inquiry and allow for broad commentary on a particular subject. The study employs a hybrid model of both structured and semi-structured approaches for the interview process to address the diverse actor groups operating within the nitrate management discourse. Structured interviews took place with agency officials, as the questions were targeted to specific points of institutional and administrative procedures. Semi-structured interviews took place with the farmers, as the aim was to elicit information about their experiences and perspectives.

The interviews took place on-site and were composed of ten to fifteen questions clustered into particular themes grounded in the theoretical framework. The study consists of nine interviews, which were recorded and averaged one hour. In addition, three written correspondences with agency

officials are included in the interview analysis. Three interviews were conducted with farmers residing within the Silesia Province. An interview with the president of an agricultural cooperative in the Silesia Province took place. Three interviews were conducted with agency officials at two different regional offices of the Agency of Restructuring and Modernization of Agriculture (ARiMR): two in Częstochowa and one in Toruń. One interview was made with an academic researcher and expert on nitrate management, who served as a former advisor to the Ministry of Environment and worked extensively with local environmental non-governmental organizations. An interview with the president of the National Council of Agricultural Chambers (KRIR) was conducted. In addition, one interview took place with an agricultural advisor and expert on the Nitrates Directive.

The analysis of the interviews centered on the pre-determined themes, which established the framework for the types of questions. The themes relate to farmer perceptions, state actor views, actor relations, and institutional structure and arrangements, in addition to roles and responsibilities. The analysis consisted of reading through each interview and manually coding text that corresponded to a particular theme. Once all of the interviews had been organized according to the thematic structure, the narrative of each theme could be analyzed. Each theme was evaluated in terms of gathering together the sections from all of the interviews that corresponded to the particular theme. In doing so, the narrative (connoted as sub-narrative), of each theme emerged. The sub-narratives collectively comprise the meta-narrative of the nitrate management discourse. An analysis of the components of each sub-narrative demonstrated correlation with particular factors, which either impede or facilitate governance capacity and, thereby, overall compliance performance. The study provides an in-depth qualitative examination of factors that impact upon implementation performance at the local level, which also reverberate and correspond to factors affecting the national level. Thereby, the study does not endeavor to be representative of Poland as a whole, but rather aims to determine if the local nitrate management discourse demonstrates convergence or divergence with the national level categorization of implementation performance.

### 3.2.3. Comparative Evaluation

The comparative evaluation of nitrate management discourses in Poland and Denmark provides a calibration in assessing how the components of political culture and governance capacity contribute to shaping compliance performance. The comparison is based upon an extensive literature review of policy documents and academic publications for each respective MS to map the nitrate management discourses. The comparison provides an assessment identifying which constraining and enabling factors contribute to implementation narratives and how such pathways demonstrate convergence or divergence in respect to a particular compliance performance categorization. Overall, the comparative evaluation serves to facilitate an understanding of how different Member States interpret EU requirements and how implementation is realized in practice.

### 3.3. Analytical Framework

The multi-method analytical approach is comprised of two components:

1. Content analysis of the interviews and literature review to identify key influencing factors,
2. Implementation discourse comparison for Poland and Denmark to determine governance capacity.

The analytical framework focuses on how the two components relate to one another to comprise the evaluation study. The flow-chart Figures 3 and 4 demonstrate the main elements and sequence of the two phases of the analytical approach. The figures provide a sequential visual of how the theoretical framework links to the separate components and how taken collectively, the analysis influences the interpretation of the results and thereby structures the discussion. The figures focus on the following: (1) factor interplay analysis (2) deducing governance capacity elements from the Poland and Denmark comparison. Together, the components comprise the framework for compliance performance assessment.

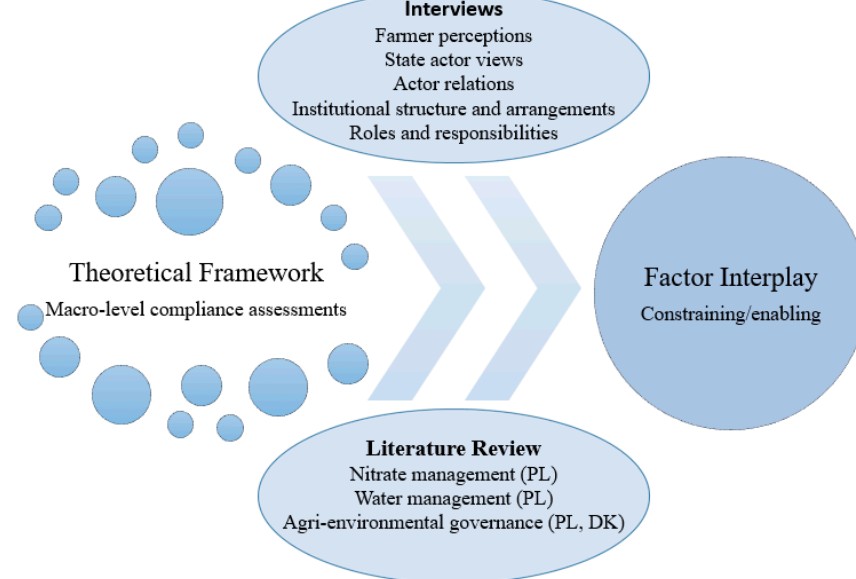

**Figure 3.** Phase 1: Factor interplay.

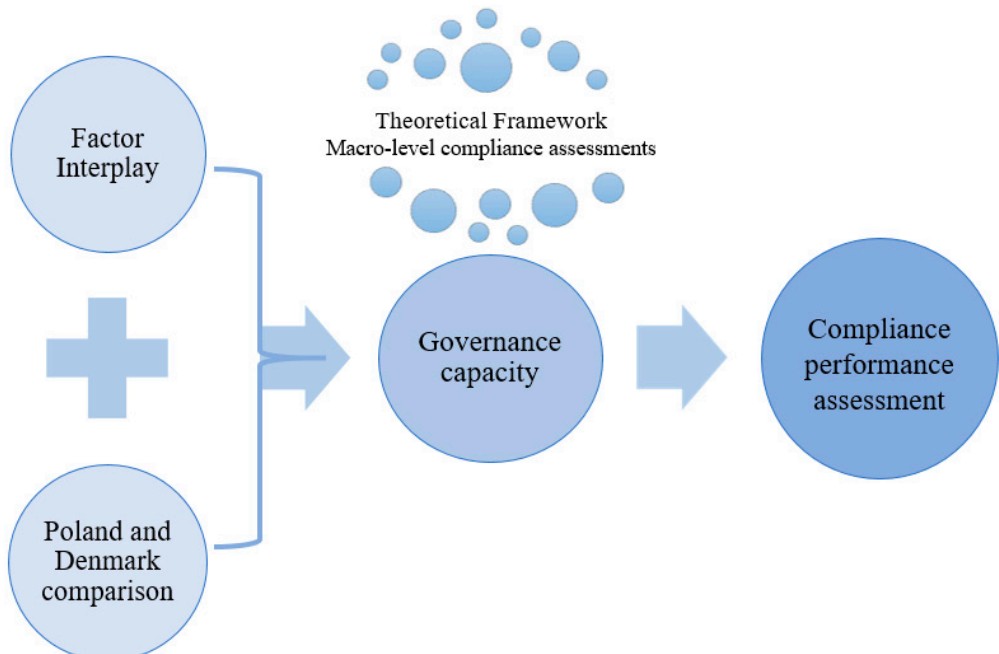

**Figure 4.** Phase 2: Compliance performance assessment.

### 3.3.1. Phase 1: Factor Interplay

The analysis of component 1 focuses on the identification of influencing factors based on the literature review and empirical evidence gathered from the qualitative research. An example of a constraining factor is low levels of social capital. Social capital refers to the levels of trust, willingness to cooperate and engage in networks of collective action [55–57]. Low levels of social capital impede efforts to implement the WFD's participatory approach, as stakeholders lack trust towards formal institutions and unwillingness to collaborate [58]. The literature review highlights diverse aspects of nitrate management that contributes to the identification of key factors influencing the overall discourse. The theoretical framework provides the lens through which the nitrate management discourse in Poland was examined within and thereby shapes the structure of the interview inquiries.

Content analysis of the interviews followed, and themes were established. The themes that emerged from the content analysis could then be assessed in relation to the factors identified a priori by the literature review. Thus, the comparison of the themes and factors act as a verification instrument to establish consistencies and inconsistencies in the results.

### 3.3.2. Phase 2: Governance Capacity

The second component of the analysis is featured in Phase 2, which contributes to confirming the presence of factors within the discourse and helps to determine at which level(s) they are operating. Figure 4 demonstrates that the composite results of the factor interplay and the Poland comparison with Denmark are evaluated in tandem to determine overall governance capacity. In doing so, a final compliance performance assessment is made.

## 4. Results

### 4.1. Implementation in Poland

The implementation of EU agri-environmental measures has been particularly problematic for the Polish state as there is a "misfit" between EU requirements and the governance capacity of the state to fulfill objectives set forth by the Nitrates and Water Framework Directives. There are various constraining and enabling factors, which inform the discourse of nitrate management and are identified in the following section. A synthesized table and a figure of the interplay are included at the end of the analysis.

The Polish agricultural sector is typified by a complex and fragmented farming system in terms of size and production, as diverse geographical conditions determine what type of farming takes place [59,60]. At the time of the study, legislative measures differed by region and even within a particular area, as specific sites were designated as NVZs [32], [61] (p. 11), [62] (p. 2). Therefore, different areas reflected different levels of engagement with nitrate management, making the discourse in Poland nuanced and varied. The variance is demonstrated by the diverse number of factors present in the interplay.

Another constraining factor is the issue of territory designation, which remains an on-going challenge for Poland. Initially, in the time period from May 2008 to April 2012, 1.48% percent of the total territory was designated as NVZ [32]. The European Commission rejected Poland's designation assessment, deemed incomplete and required that the state reform its action plans [33]. Poland increased the total designated area to 4.46% of the territory, reflected in the updated plans from 2012 to 2016 [32]. In an independent assessment commissioned by the European Union, a recommendation was made that Poland should designate the entire territory as NVZ [63]. When agency interviewees were presented with the enquiry regarding if the total NVZ designation area was sufficient, all stated that the designation level was too low. Further, there was consensus among the interviewee responses that the percentage of designated areas should be raised, but not cover the entire area. The common opinion expressed was that designation of the whole territory would be excessive and therefore, not necessary. Concern was expressed that the heavy policy load would entail significant financial investment and constraints placed upon farmers, which would lead to detrimental effects on the functioning of the Polish agricultural sector, including increased costs of production and of Polish products [61] (p. 2), [64] (p. 6).

"For sure we don't agree to cover the whole country as a whole area, but there could be some compromises so we designate, maybe not 12, but perhaps 20 percent. Yet, still the question will be if the real investments are going to this area and if the farmers are really implementing all of the requirements, all of the needed activity or not because it will not be controlled" [64] (p. 6).

In the Silesia Province, there are 53,000 farms [61] (p. 8). There is a lack of oversight and monitoring, due to the large number of small, self-sufficient farms that largely do not complete fertilization plans, which makes an assessment of the inventory of total nitrogen use difficult. The historical legacy of Soviet occupation and imposed communist rule contributes to a public culture that is suspicious of government

control, as evidenced by the low levels of social capital [58], [62] (p. 16), [64] (p. 8), [65,66], [67] (p. 2). Approximately 25% of the population resides in the countryside and farmers are treated as a special interest group that political parties cater to [64] (p. 5). The political situation is such that farmer's exercise considerable influence as a voter constituency and a pro-farmer attitude prevails in the public sphere. Therefore, there is a lack of political will to control how measures are implemented by farmers [62] (p. 9), [64] (pp. 2, 5). The lack of political will extends further, as the government seeks to protect farmers from European standards [64] (pp. 2, 4, 5, 7). The immediate consequence of the lack of enforcement is that there is a discrepancy among formal compliance and local implementation—what is written in official documents does not match what is happening in practice on the fields at local level [68] (p. 2).

It is important to understand the motivations and perceptions which contribute to the mindset of a particular stakeholder group, including farmers, as different perspectives lead to variance in the interpretation and, thereby, realization of policy in practice. In the case of Poland, the historical legacy of an imposed totalitarian system is viewed as a constraining factor informing the domestic situation [58,69]. Current institutional arrangements and administrative culture reflect the top-down, centralized governance culture of the past communist regime [66,70–73]. Under the totalitarian regime, the market was isolated by protectionist measures. Once the domestic market was liberalized and access opened to international markets, farmers had to contend with competition and low prices, which they were not prepared for. Polish farmers lacked significant capacity in terms of knowledge and technical expertise [68] (p. 3), [69,74]. Additionally, in joining the EU, the state experienced a situation of policy overload [71]. The conditions in which the sudden institutional and economic changes were taking place were entirely unfamiliar and to this day, are characterized by uncertainty, due to continued frequent changes taking place [67] (p. 4), [75] (p. 5), [76] (p. 4).

The heavy load of performance expectations and policy requirements imposed upon farmers amidst weak capacity to keep pace with the changes exacerbates pressure on the already burdensome nature of farming as a profession. Farming is fraught with uncertainties and risks inherent to the system, such as the instability of commodity prices, unpredictable weather affecting production, and a heavy policy requirement load [62] (p. 22), [75] (pp. 5, 8), [76] (p. 5). Furthermore, the role and responsibilities of farmers is changing against the backdrop of payments being coupled with environmental measures [67] (p. 4).

The group of Polish farmers interviewed for the study expressed that there should be more support for farming. as the EU increases expectations of farmers. Indeed, the shifting role and greater responsibility of farmers is expressed in policy documents [77]. The role of farmers is evolving to that of nature managers in addition to food producers [31]. Polish farmers interviewed were in agreement that environmental considerations are important in relation to the ecosystem services provided to farming and that measures should be taken to protect the environment. Yet, it was also expressed that farmers hold a self-perception of playing a significant role in providing the vital service of food production on behalf of society. The farmer interviewees expressed that the costs of environmental protection should be borne by all of society. Thus, there should be more support for farmers to comply with agri-environmental measures [62] (p. 4) [78].

The issue of environmental protection is complex, due to mixed perceptions. EU environmental measures are viewed as a barrier for economic growth and development [64] (p. 6), [71,79]. Behavior exhibiting a lack of willingness and even resistance to compliance has been observed when environmental directives are perceived as an impediment to the advancement of Polish economic interests. Existing environmental regulations are exogenous in terms of originating as a political mandate from a higher level (European Union) which further deepens the distrust Poland holds. The origin of distrust was pre-existing to accession, as the EU is viewed as an external actor, which seeks to influence the internal affairs of the Polish state. The pre-existing political culture, which holds a negative perception of environmental protection coupled with distrust towards the EU as an external institution contributes to a skeptical stance in terms of compliance with environmental EU directives.

Following the completion of this study, the Polish government adopted a new approach to nitrate management, due to litigation pressure from the European Commission on the basis of insufficient implementation. In response, Poland implemented a uniform approach in designating the entire territory as an NVZ. A new Water Act enacted in 2017 aims to better address water pollution from diffuse sources through the development of more targeted Nitrate Action Programs [80]. Based on the results of the present study, implementation of the more ambitious strategy will likely entail challenges for Poland.

*4.2. Influencing Implementation Factors*

The analysis resulted in the identification of 19 factors: 13 constraining (Table 2) and 6 enabling (Table 3). The factors are not independent of one another, but rather coalesce as an interplay, which establishes the discourse that determines governance capacity and ultimately, implementation performance.

The findings from the analysis identify a number of constraining factors that limit the governance capacity of Poland to comply with implementation requirements of the Nitrates Directive. Table 2 lists the 13 constraining factors identified from the interviews, along with the extensive literature and policy review. Three categorical distinctions are given for each of the factors based on (1) the descriptive category of cultural, political and administrative, (2) the operational level at which the factor is present of societal, national, regional local and interest group levels, and (3) the source of the pressure being either external or internal to the state. As the results from Table 2 demonstrate, the constraining factors are diverse and collectively illustrate the complexity of the confluence of factors that impact upon and shape governance capacity. In the case of Poland, the majority of the constraining factors constitute internal pressures, thereby signaling that actions oriented towards the domestic situation are most pertinent. The analysis reveals that socio-cultural factors underlie many of the constraining factors shaping the implementation discourse in Poland. Overall, Poland experiences difficulty in compliance with requirements of the Nitrates Directive, leading to a convergence of a laggard categorization exhibited at both the local level of the empirical case study and national level.

Table 3 provides an overview of the six enabling factors identified in the analysis. A discussion of the implications of the interplay of enabling and constraining factors is beyond the scope of this study. It is important to note that the enabling factors could be useful to consider in shaping future policy considerations and for further research.

**Table 2.** Constraining factors identified in component 1 of analysis for the evaluation of governance capacity.

| Influencing Factor Constraining | Explanation | References | Descriptive Category Cultural Political Administrative | Operational Level Societal (S) National (N) Regional (R) Local (L) Interest Group(I) | Pressure Internal (IN) External (EX) |
|---|---|---|---|---|---|
| Negative environmental perception | Environmental protection viewed as a barrier to economic development. Lack of knowledge and interest lead to stagnation of integrating environmental considerations in sectors | [68,71,79,81] | Cultural | S | IN EX |
| EU acts as external, centralized, regulatory institution | EU perceived as regulatory supranational institution imposing its interests | [70,71] | Cultural Political | S | IN EX |
| Historical political tradition of imposed system | Imposed totalitarian communist regime governed top-down | [58,66,71,82] | Cultural | S | IN |
| Low levels of social capital | Lack of general trust contributes to a weak capacity of civil society to engage and constitutes mental barrier | [58,66,69–71] | Cultural | S | IN |
| Resistance to external norms: nitrate pollution attributed to agricultural sources | Viewed as a local problem, rather than a national priority. More emphasis placed on sewage treatment or control of industrial pollution | [69,79,81] | Political | N | IN EX |
| Diffusion and mismatch of responsibilities amongst various agencies leading to lack of cooperation | Inter-sectoral integration difficult to achieve and thereby limits the efficacy of fulfilling EU Directive objectives | [66,69,71,73,79,81] | Political Administrative | N, R, L | IN |
| Financial constraints | High costs associated with measures to implement such as building manure storage containers | [68,71,74,79,81] | Political | N, R, L | IN |
| Command-and-control approach | In terms of national level setting policy priorities, which does not account for the variance in policy priorities of the provinces, leading to the lack of coherent policy objectives | [69–73,79] | Cultural Political | N | IN |
| Competing policy priorities | A variance of priorities present at different levels of government and in different provinces. Production interests and economic growth prevail | [71,79,81] | Political | N, R, L | IN |
| Lack of resources | Financial and other competencies such as proper amount of staff and sufficient levels of knowledge | [69–71,74,79,81] | Administrative | N, R, L | IN |
| Heavy policy load | Adoption of entire EU body of law upon accession | [49,71] | Administrative | N, R, L | EX |
| Fragmented farming system | Variance in size of farms and production taking place, along with large number of farms | [64] (p. 2) [83] (p. 1) | Administrative | N, R, L | IN |
| Relative power of farming constituency | The large number of farmers together make a influencing agricultural policy priorities | [61] (p. 9) [64] (pp. 2, 5–7) | Political | I | IN |

Table 3. Enabling factors.

| Influencing Factor: Enabling | Explanation | References | Descriptive Category Cultural Political Administrative | Operational Level Societal (S) National (N) Regional (R) Local (L) Interest Group (I) | Pressure Internal (IN) External (EX) |
|---|---|---|---|---|---|
| Legislative framework provided | EU provides framework to address non-point source pollution stemming from agricultural sources in the form of the ND, WFD, CAP, etc. | [1,2] | Political | N, R, L | EX |
| Degree of devolution of power | From national to local levels with reinstatement of local governments and assignment of competencies and responsibilities | [72,73] | Administrative | L | IN |
| Compliance pressure | Pressure to comply with directives by EU and instruments invoked to ensure sufficient implementation | [33–35] [61] (p. 7) | Political | N, R, L | EX |
| Prior examples of policy fit | Previous water management system based on river basins as WFD requires | [84] | Administrative | R | IN |
| Diverse communication channel system | Agencies and farmers use a diverse array of communication channels to access and provide information and knowledge, contributing to a shared understanding | [61] (pp. 1, 3, 4) [75] (p. 6) [76] (pp. 1–3) | Cultural Administrative | S | IN |
| Greater acceptance and legitimacy of nitrate pollution | General acceptance that nitrate pollution stems from agriculture and that environmental measures are necessary to address the issue | [62] (pp. 4, 5) [68] (pp. 2, 4) | Political | S | IN |

A greater number of constraining factors were identified in comparison to the number of enabling factors, which collectively constitute a descriptive categorization of the nitrate management discourse. The main difference between Tables 2 and 3 centers on the lack of cultural enabling factors, while a number of constraining factors are cultural. The dissonance demonstrates the important role that culture plays in the interpretation of policy, which is imbued with normative values and cultural understandings and thereby shapes implementation outcomes. In the case of Poland, the country did not partake in constructing the EU environmental acquis, which constitutes the legal framework guiding MS behavior of natural resource management and how management ought to be realized as social practice. Nitrate management is a complex challenge for Poland, requiring immense resources, as the agriculture sector navigates structural transitions amid a shifting agri-environmental policy landscape. The multi-scalar and rapid speed of the transition exerts significant pressure in being able to address diffuse pollution.

Interplay of Constraining Factors

Assessing the interplay of constraining factors that shape the nitrate management discourse leads to an understanding of the interactions shaping Poland's position in being able to address nitrate pollution stemming from agricultural sources. Figure 5 illustrates the constraining factor interplay operating at different institutional levels within the domestic context of Nitrates Directive implementation of the empirical investigation. The results are based on the interviews with advisory agency officials, farmers, and an academic expert, in addition to the written correspondences with

government agencies that constitute the discourse actors involved in implementation performance. The figure demonstrates the constraining factor of low social capital operates across institutional levels and more broadly, within society, which weakens governance capacity and impedes overall compliance performance. Social capital is an important influencing factor shaping the ability of Member States to implement the Water Framework Directive, as the organizational structure of institutions and coordination of actions to manage water resources ultimately rests on the success of stakeholder interactions. The implications of Figure 5 stipulate a dialectic relation of low social capital levels that perpetuates a negative feedback loop. Low social capital exerts internal pressure within the domestic context, as well as shaping the perception and response to external influences. In tandem, the external influences of the historical legacy of occupation and pressure from the European Union to comply with environmental directives contribute to lowering social capital.

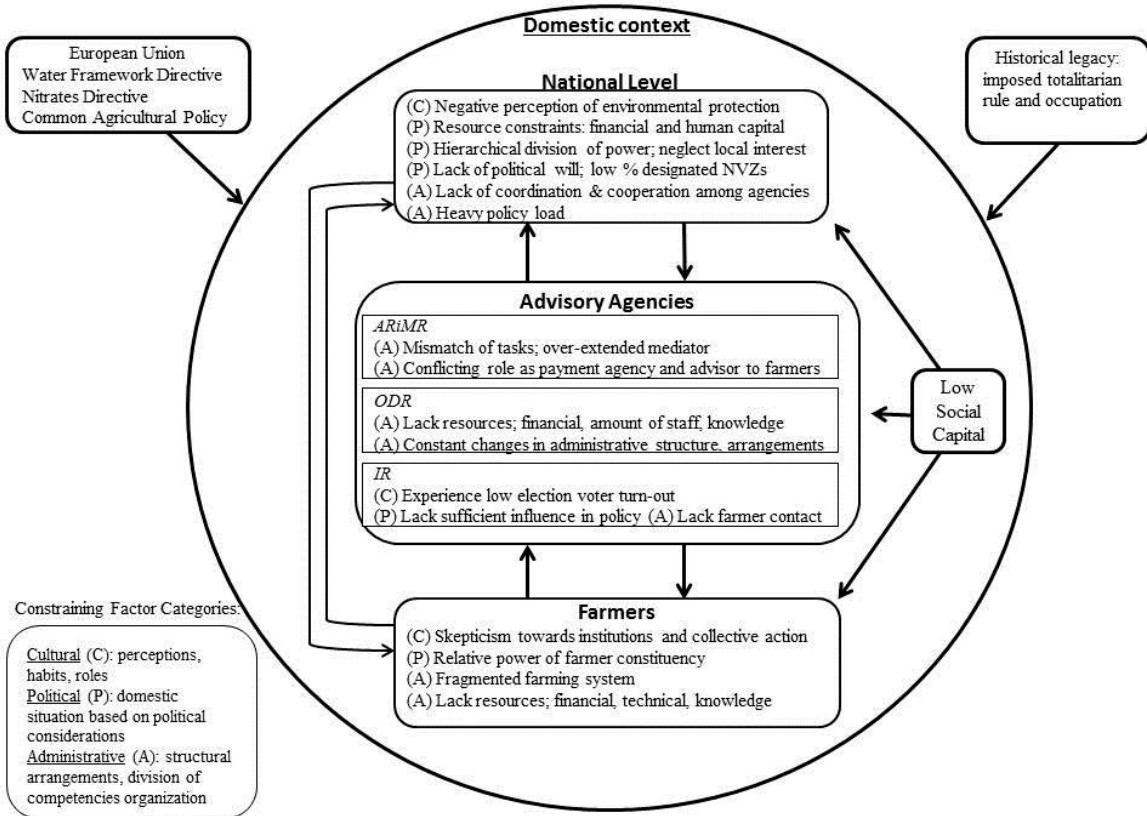

**Figure 5.** Interplay of the constraining factors that shape the nitrate management discourse.

### 4.3. Denmark's Shifting Implementation Performance

Denmark has historically demonstrated a "pioneering" role [41], in being one of the first states to recognize and address diffuse pollution stemming from agricultural sources. Denmark exhibited leadership in setting ambitious nitrate pollution reduction targets and policies before the establishment of the Nitrates Directive [85]. Aquatic plans had already been implemented before the ND and WFD were enacted, with Aquatic Environmental Action Plan I, II, and III in 1987, 1998, and 2004. In the broad body of literature ranking EU Member States on their environmental policy performance, Denmark is consistently ranked as a leader [41]. Yet, while Denmark has taken considerable measures to improve N-efficiency and to reduce both N-surplus and environmental loadings, further reduction measures are needed to meet compliance requirements of the Water Framework Directive [86]. Denmark's performance in regards to nitrate management originally placed the state in the position of environmental leader, [42,87,88], yet the state's performance is lagging in recent years, as illustrated by the difficulties in implementing the Water Framework Directive [22,89,90].

The 2009 deadline of River Basin Management Plans (RBMPs) was delivered in 2014, a delay of five years. The plans are essential to establish measures responsible for policy deliverables. Failure to make and submit plans effectively stalls the practical implementation of the WFD. When plans were submitted by Denmark in 2012, they had to be rescinded because the public consultation period had been too short and thereby violated domestic law. The delay in the submission of RBMPs led to an infringement case brought against Denmark by the European Commission in 2014. Denmark's behavior of foot-dragging in failing to adopt the WFD in a timely and correct manner is attributed to a multiplicity of intersecting biophysical and governance challenges that constitute constraining factors. The factors include high livestock density, a large proportion of agricultural land area with many fields in rotation, an 8750 km long coastline with many shallow nitrogen vulnerable estuaries, combined with pressures from increased rainfall and temperatures related to climate change. In addition, the governance system in Denmark undertook significant institutional structural changes in 2007 by abolishing counties and establishing new municipalities. Former responsibilities of water planning, water quality, and environmental protection were devolved to the counties. The organizational changes from a municipality-county-state structure to a municipality-region-state structure, effectively consolidated governance arrangements, leading to a centralized water management system. The consequence of the changes in governance arrangements resulted in a misfit in being able to realize an RBMP approach with stakeholder participation. [89].

The Environmental Ministry began the process with an open dialogue and plans for how to succeed with the implementation of the RBMPs in a timely manner. The first RBMP cycle began with a public consultation phase that urged municipalities, regions, citizens, and interest organizations to develop ideas and suggestions for how to implement the WFD in Denmark. The responsible authority (Nature Agency under the Environmental Ministry) received approximately 2500 contributions from a wide range of stakeholders. The original plan stipulated that the contributions from the public consultation would result in a White Paper on implementation strategies. However, due to time constraints and limited resources, the municipalities were only provided with general feedback from the consultation. The resultant RBMP working program thereby included little from the public consultation process [91].

In addition to the formal delays in compliance with EU regulations, there is a lack of consensus among key stakeholders regarding further reduction targets and appropriate implementation measures. The validity of present scientific data has been questioned, along with the perception that there has been too narrow a focus on N-discharge and an over-implementation of environmental policy targets [18,90]. The views support the narrative of domestic protectionism in terms of safeguarding the economic interests of Danish farmers, in comparison to competing countries of export markets, which have also not complied with the EU environmental policies [92]. The situation was further impeded by economic crisis and a high reliance on export products compared to neighboring countries (approximately 2/3 of the production value).

The protectionist narrative contrasts with the previous patterns of implementation behavior exhibited. Denmark previously demonstrated high ambitions for environmental protection, a culture of compliance, leadership, and a commitment to delivering upon policy objectives. Yet, Denmark appears to have taken a step back from its role as an environmental leader and adopted a minimalist position in relation to implementation of the WFD. However, this also coincides with a shift in policy from a general regulation, where the same reduction targets were set for all of Denmark, to a geographically targeted regulation. In addition to the ND measures, differentiated reduction targets and implementation measures are set for each watershed, in accordance with the WFD [93,94]. The argument is that this shift takes time to implement and for intended effects to materialize in targeted watersheds. Yet, Denmark has lowered its overall environmental ambition, partially in response to the fact that many fellow Member States failed to deliver results at the same level. Therefore, during the period of WFD implementation, the Danish case demonstrates environmental laggard behavior, due to a minimal and delayed implementation response, albeit from a differentiated position than other EU

laggards. The competitive advantage of being a leader in achieving environmental targets is lost if the majority of Member States fail to perform at the same or similar level. Achieving EU policy deliverables requires every Member State's commitment and contribution. The WFD's objective of achieving good status of all waters cannot be met without Member States acting collectively to manage water as a common-pool resource [17].

Part of the explanation accounting for the significant change in Danish policy may be related to the increasing marginal cost of reducing nitrogen (N) loads from agriculture. When the first legislation aiming at reducing N loads was adopted in the 1980s, agricultural practice was not optimized in relation to nitrogen use efficiency and it was possible to significantly reduce N loads with minor adverse impacts on crop yield and farm economy. By redefining good agricultural practice and regulating fertilizer applications, the N load from agriculture was reduced by 50% over two decades [86]. The nitrogen (N) reduction targets adopted in connection with the first WFD River Basin Management Plans scheduled for 2009 stipulated an additional substantial reduction, following the 50% reduction achieved between 1985 and 2003 as part of the ND. Because all the low hanging fruits had already been harvested, the additional reduction would imply costly measures and was thereby assessed as a major threat to farmers. As WFD implementation was furthermore attempted to be implemented by a rather centralistic approach without active stakeholder involvement, the implementation met heavy resistance, including a radicalization of farmer organizations. At the same time, farmers could argue with some justification that Denmark was over-implementing compared to neighboring countries who had protected their farmers from regulations similar to those already existing in Denmark. Altogether, these factors caused the WFD implementation to become politically difficult. Thus, the N load has not been reduced significantly since 2005. Yet, in recent years, some discussions and pilot experiments have been made to actively involve farmers in identifying locally based solutions to improve the WFD implementation [91,95,96].

The Danish case is difficult to typify due to constraining elements unique to the national context, along with differentiated responsibility in terms of the variance of nitrate pollution pressure. Due to the constraining elements of intensive agriculture and vulnerable bodies of water, Denmark has a greater nitrate pollution pressure in comparison with other Member States [82]. Therefore, greater measures are needed to fulfill the requirements for reaching good ecological status of waters and to meet EU obligations of the WFD. Yet, there is a perceived risk of reduced economic returns and being placed at a competitive disadvantage if further N reductions are made. The total cost of the investment of financial resources and other services, along with economic losses, is viewed as outweighing the environmental benefits. Denmark is placed in a difficult position as the state seeks to balance its domestic interests, which are at a dissonance with its commitment to EU water policy goals. Denmark's interest is to remain a global leader in the export of livestock, yet its ability to deliver upon EU water quality requirements is compromised, leading to a conflict of interests. The role of Denmark is shifting as the state responds to dynamics taking place both within the national discourse and the larger geopolitical landscape the country is situated within. Denmark is predicted to continue with foot-dragging behavior as domestic economic interests conflict with EU objectives. The European Commission will most likely pressure Denmark to raise the state's level of performance. The reality of the domestic situation is difficult to reconcile with EU expectations. In doing so, the validity of such expectations may be challenged and could even lead to reform of water policy itself to better match the national contexts of Member States. However, one point that may lead to an improvement of the situation is the high level of ambition and investments in measures to mitigate greenhouse gas (GHG) emissions, which may deliver synergistic benefits of reduced nitrogen pollution in the agricultural sector [97].

## 5. Discussion

*5.1. MS Compliance Comparison*

A realization emerges that despite EU environmental directives establishing common objectives and norms of environmental standards, the share of responsibility varies considerably by MS. In other words, some MS may be required to do more relative to other states to achieve the overarching EU targets. In the case of both Poland and Denmark, there is considerable pressure to reduce nitrate pollution stemming from agricultural sources. Poland is the sixth largest country in the European Union with a total area of 312,679 km$^2$ and has a population of 38.1 million [31]. Poland makes up half of the total population of the Baltic Sea Basin. Further, 98% of its surface waters drain to the Baltic Sea [78]. Based on these conditions, Poland exerts significant influence on the ecological status of the Baltic Sea [98]. Denmark is approximately one-seventh the size of Poland, with a total area of 42,925 km, a population of 5.8 million, and all of its surface waters are drained to the surrounding aquatic environment. Agriculture makes up 60% of the total land area use and the majority is intensive production, focused on livestock for export, which leads to high amounts of nitrate loads produced [99–103]. For both states, nitrate management is situated within a context of a strong agrarian cultural tradition, which contributes to agriculture playing a significant role in the interests of each respective state.

Poland and Denmark face considerable constraints in terms of managing nitrate pollution stemming from agricultural sources. Confronting the national contexts of both states highlights the challenge of achieving higher water quality standards for Europe as stipulated in the WFD. A comparative analysis of the nitrate management discourses presents pertinent inquiries with regards to the feasibility and overall efficacy of EU environmental policies: are EU targets at a dissonance with the national contexts of Member States? Further, how are directives to account for differentiated responsibilities in terms of reduction targets? Furthermore, it is important to take into account that Denmark and Poland vary considerably in their interpretations of EU environmental policy, which affects how measures are applied and carried out in practice.

### 5.1.1. Political Culture

A comparison of the political cultures in Poland and Denmark demonstrates considerable divergences, which influences state behavior in relation to how nitrate pollution is perceived and acted upon. Poland has sought to minimize the issue to privilege domestic interests over EU environmental protection priorities. Negative perceptions of environmental protection, skepticism towards the EU, foot-dragging on NVZ territory designation, and the high cost associated with implementation measures of the Nitrates Directive contribute to a constricted political culture.

Denmark recognized nitrate pollution as an issue before being formally addressed by the EU. Prioritization of the issue at the national level established a political culture where nitrate management was a salient issue. The state was well-organized and allocated resources for implementation, supported by a political culture of compliance. Yet the past 20 years demonstrate a drawback from earlier ambitious targets and a narrative emerge of "over-implementation" demonstrated by the downscaling of environmental targets and ambitions.

### 5.1.2. Governance Capacity

The results of the analysis demonstrate that policy approaches shape perceptions, practices, and the behavior of actors at all levels of nitrate management discourses. Based on the comparative theoretical framework of compliance performance assessments, Poland aligns with the categorization of misfit [24], concerning environmental compliance performance. Despite the recent policy reform of entire territory designation as an NVZ, Poland's highly fragmented farming system requires a diverse array of stakeholders to cooperate and coordination of institutional arrangements spanning various sectors operating at different governance levels. Where European Union regulations have

left space for national provisions, EU measures have rarely been fully transposed, giving rise to competing interpretations among different stakeholders and public authorities [65]. Implementation of EU environmental directives has thereby required significant financial resources and institutional restructuring to manage the significant policy load. An examination of the situation in Poland reveals that nitrate management is a contentious socio-political issue, with culture playing an important contributory role in shaping the discourse and policy approaches. Ultimately, the constraining factors present in the discourse exert considerable pressure on Poland's governance capacity to effectively address agriculturally induced nitrate pollution and constitutes a constricted political culture.

While Poland has put in place regulatory and institutional frameworks that are relevant to the formal compliance measures of the Nitrates and Water Framework Directives, the state has not succeeded in fulfilling the reduction targets of nitrate effluent loads to the Baltic Sea. The inability to meet reduction targets is largely due to the confluence of constraining factors that limit the ability of state and non-state actors to adequately address agriculturally induced nitrate pollution. The various pressures are interlinked and together contribute to a situation where the nitrate management discourse faces challenges in terms of Poland having sufficient governance capacity to fulfill its Member State obligations. The lack of sufficient governance capacity contributes to a "misfit" categorization of environmental compliance performance.

In the case of Denmark, there are a number of enabling factors that advance governance capacity and contribute to an implementation performance categorized as "front-runner" and "maximalist". Yet, it is important to acknowledge that general categorizations may not hold when evaluating performance of a particular policy subfield. Further, rankings are subject to temporal and spatial variation, as policy development and outcomes are continuously evolving in response to the current context [41]. In the case of Denmark, past environmental performance regarding nitrate management has demonstrated a leader role within the EU. At present, the Danish status is more difficult to rank based on laggard performance of the WFD. An evaluation of the nitrate management discourse demonstrates that environmental efforts have stalled. While, Denmark was a front-runner in terms of reducing N loading and optimizing N use efficiency, the state has now adopted a foot-dragging position in making further reductions. The recent trends demonstrate a change in approach, reflecting a narrative of over-implementation, expressed by the downscaling of environmental targets and ambitions.

The narrative of over-implementation of EU environmental protection measures emerged as evidence of compliance records of other Member States demonstrated weak performance and minimalist transposition of measures. Despite Denmark making considerable progress in reducing pollution from diffuse agriculture sources, overriding productionist concerns combined with weak compliance of other EU states have dampened Denmark's interest to continue as an environmental leader. Coupled with the fact that 60% of the entire Danish territory is dedicated to agricultural production, there appears to be a political limit that has been reached, met with an unwillingness to set more ambitious targets. In doing so, Denmark's environmental leader role is questioned in terms of its saliency and applicability to typifying the Danish case.

*5.2. Implications*

Agriculturally induced water pollution is part of the larger global challenge of unsustainable resource management, which at its core is a crisis of governance. Further, research demonstrates that half of the global population expresses distrust in government institutions [104]. This governance gap persists worldwide and applies to the management of natural resources as well. Natural resources are shared public goods entailing collective action in designing systems of management regarding their use, accessibility, and distribution. Widespread environmental degradation and dwindling levels of trust in traditional institutions are symptoms demonstrating that prevailing governance systems are inadequate and in need of reform. Water-related challenges contain opportunities for addressing unsustainable resource management by providing the space for new modes of governance based on collective action to be established.

Co-governance may be an effective framework in bridging the existing gap. The co-governance approach entails the pursuit of better governance based upon an understanding of the inherent complexity of environmental challenges and is reflected in tailored polices that incorporate meaningful stakeholder engagement in decision-making processes. Local stakeholder engagement and participatory decision-making are the main working components demonstrating effectiveness in building relations of trust among actorsand institutions to deliver long-term success. Vested interest is established when local actors are involved as an integral part of a project or policy. The advantages of promoting participatory processes are to improve governmental accountability, build trust and mutual understanding, empower stakeholders to take ownership and responsibility for the provision of public services along with efficiency gains [105].

A key component of the WFD is implementing a participatory approach, in recognition that public engagement is fundamental for an integrated water management approach to be realized in practice and to deliver upon environmental policy objectives [106,107]. Indeed, research has established that long-term sustainable water management requires cooperation through collaborative governance arrangements [95,108–110], demonstrating the significance of social dynamics in achieving improved environmental outcomes. Actors operating at different levels (national, regional, and local), are required to self-organize, define roles, delegate responsibilities, develop context appropriate strategies, coordinate communication, and, ultimately, deliver policy implementation outcomes. As the comparative study illustrates, implementation performance constitutes an interplay of influencing factors amid a shifting policy landscape. Thus, realizing a co-governance system depends on fostering enabling conditions of governance capacity that fit a particular context. In the case of managing agricultural diffuse pollution, different degrees of "fit" between top-down and bottom-up social organizational arrangements are necessary to address the differentiated socio-institutional settings of MS. Therefore, co-governance may serve as a means to improve water resource management, while simultaneously bridging the governance gap [91]. Agriculture is central to sustainable water management and situated at an inflection point to create new modes of governance that address water-related challenges.

## 6. Conclusions

At present, both Poland and Denmark demonstrate laggard implementation performance of the WFD, despite differentiated nitrate management discourse trajectories. Both MS are under considerable pressure to reduce nutrient loadings to the Baltic Sea and are situated within a political landscape where domestic interests conflict with EU ambitions for water protection. The situation looks different when evaluating the implementation performance of the ND, as discussed above with regards to Denmark's performance. In that case, Denmark is a leader in achieving N reduction targets under the ND, but exhibits a foot-dragging position in failing to fulfill the more ambitious requirements of the WFD. For Poland, fragmentation at political, administrative and cultural levels constrains governance capacity in delivering N reduction targets under the ND and subsequently, the WFD, leading to an overall laggard position.

The results of the comparative evaluation demonstrate the nuances of differentiated MS contexts that constitute nitrate management discourses. A complex interplay of factors influences MS governance capacity to manage nitrate pollution stemming from agricultural sources and comply with EU environmental directives. Despite Poland and Denmark representing significant divergences in management discourses, both MS face considerable constraints in being able to achieve the WFD target of good ecological status of waters by the 2027 deadline.

Agricultural management is situated at an inflection point where a trifecta of multiplicity governance interactions (multi-actor, multi-sector, and multi-level, as defined by Liefferink et al. [48]), intersect and culminate in water governance outcomes. The issue of agricultural diffuse pollution illustrates the complexity of water governance as an evolving policy landscape shaped by a confluence of influencing factors affecting the overall ability of MS to comply with the WFD. The present study

identified governance capacity factors that impact upon governance arrangements and structure implementation discourses, which thereby drive compliance outcomes. The results demonstrate that implementation performance constitutes a dialectic process of accommodating, adjusting, and contesting EU policies within domestic MS contexts.

While the comparative evaluation centers on Poland and Denmark, the study is illustrative of the broader trend of MS struggling to comply with the WFD. Further comparative research into how factor interplay influences implementation outcomes of MS is necessary to be able to draw learning lessons across the EU. In particular, more comparative studies between old and new MS can contribute towards the development of a coherent understanding of managing the wicked water management challenge of agricultural diffuse pollution.

Understanding the intricate interplay of factors that contribute to governance capacity aids in identifying the root causes underlying the persistent structural barrier of fragmentation of governance arrangements. Building governance arrangements based on a co-governance approach of public participation can potentially initiate new social–institutional settings to realize an integrated water governance system. A water governance architecture that actively promotes horizontal and vertical integration through institutional and administrative coordination, along with stakeholder collaboration, can support an enabling framework to achieve the WFD objectives. Thereby, locally tailored and differentiated policy approaches can better target agricultural diffuse pollution. Building a more appropriate water governance system based on a co-governance model that accounts for the complexity inherent in diffuse pollution can enhance governance capacity and bolster MS compliance. High levels of compliance serve to uphold the integrity of the environmental acquis and advance the environmental interests of the EU. Overall, the result strengthens the legitimacy and efficacy of the EU in its efforts to realize an integrated water governance system for the sustainable management of Europe's waters.

**Author Contributions:** Conceptualization, E.N.P.; methodology, E.N.P.; formal analysis, E.N.P.; writing—original draft preparation, E.N.P.; writing—review and editing, E.N.P., M.G., J.C.R. and T.D.; supervision, M.G., T.D., J.C.R. All authors have read and agreed to the published version of the manuscript.

**Funding:** This research received no external funding.

**Acknowledgments:** Maps in this article were created using ArcGIS® software by Esri. ArcGIS® and ArcMap™ are the intellectual property of Esri and are used herein under license. Copyright © Esri. All rights reserved. For more information about Esri®software, please visit www.esri.com.

**Conflicts of Interest:** The authors declare no conflict of interest.

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
