# Peer review of "Nitrate Management Discourses in Poland and Denmark—Laggards or Leaders in Water Quality Protection?"

_water, doi:10.3390/w12092371_

Round 1

Reviewer 1 Report

This is a good social science analysis of an important policy issue and is nearly publishable as submitted. Some minor revision is needed, however. In the Introduction (lines 60-65), the authors need to explain in greater detail what they mean by these states being "laggards" in respect to this issue - laggard compared to other water management issues? With regard to pollution compliance generally? And most important, in comparison to other EU states performance? (a point I will come back to momentarily). On line 83 and ff, I am not convinced by the authors' definition and characterization of "governance capacity." They would do well to review some of the more theoretical literature on state capacity and "state thickness" (e.g., Theda Skocpol, Nelson Lichtenstein) to better account for other variables central to capacity including size and training of he bureaucracy, ability to inflict costly penalties, precision of regulation, etc.

I al am not clearly convinced about their sampling methods and interviewee conclusions -- I'm not so much troubled by the small samples as I am by lingering questions unanswered in the manuscript.  Particularly, one would like to know: how were these individuals selected for interview? What were their precise roles? Why do they feel this sample constitutes a sufficient number of representative officials/academics from which to draw national level conclusions?

Lastly, in the Conclusions (lines 725 ff), the authors need to expend their assessment of state performance on the nitrate loading pollution issue more broadly. They have assessed only two states: have we any reason to believe these states are outliers relative to the rest of Europe? Is any EU member state meeting its nitrate reduction targets well? Even if this question cannot be decisively answered, the journal's readership will want to know the extent to which conditions in :Poland and Denmark; resemble those in other EU states; can be said to differ in significant ways; and/or are instructive of broader challenges facing all states with regard to this wicked water quality problem.

Reviewer 2 Report

see pfd please
